# Research Progress on Cannabinoids in *Cannabis* (*Cannabis sativa* L.) in China

**DOI:** 10.3390/molecules28093806

**Published:** 2023-04-29

**Authors:** Xiangping Sun

**Affiliations:** Institute of Bast Fiber Crops, Chinese Academy of Agricultural Sciences, Changsha 410205, China; sunxiangping@caas.cn

**Keywords:** cannabis sativa, cannabinoid, cash crop, terpenoid phenolic compounds, China

## Abstract

Cannabis (*Cannabis sativa* L.) is an ancient cultivated plant that contains less than 0.3% tetrahydrocannabinol (THC). It is widely utilized at home and abroad and is an economic crop with great development and utilization value. There are 31 countries legalizing industrial cannabis cultivation. Cannabis fiber has been used for textile production in China for 6000 years. China is the largest producer and exporter of cannabis. China may still play a leading role in the production of cannabis fiber. China has a long history of cannabis cultivation and rich germplasm resources. Yunnan, Heilongjiang, and Jilin are three Chinese provinces where industrial cannabis can be grown legally. Cannabinoids are terpenoid phenolic compounds produced during the growth, and which development of cannabis and are found in the glandular hairs of female flowers at anthesis. They are the active chemical components in the cannabis plant and the main components of cannabis that exert pharmacological activity. At the same time, research in China on the use of cannabis in the food industry has shown that industrial cannabis oil contains 13–20% oleic acid, 40–60% omega-6 linoleic acid, and 15–30% omega-3 α-linolenic acid. At present, more than 100 cannabinoids have been identified and analyzed in China, among which phenolic compounds are the main research objects. For instance, phenolic substances represented by cannabidiol (CBD) have rich pharmacological effects. There are still relatively little research on cannabinoids, and a comprehensive introduction to research progress in this area is needed. This paper reviews domestic and foreign research progress on cannabinoids in cannabis sativa, which is expected to support cannabis-related research and development.

## 1. Introduction

Cannabis (*Cannabis sativa* L.) is an ancient cultivated plant and an economic crop with great development and utilization value [1]. It has been used for thousands of years, primarily as a source of stem fiber (both the plant and the fiber termed “hemp”) and as a source of edible seeds [2,3]. Cannabis fiber has been used for textile production for 6000 years in China. It is cultivated (or found wild) all over the world and is mainly distributed in Asia, especially in China. Most scientific investigations authorized in Western countries were either forensic studies to aid law enforcement, or medical and social research specifically intended to document and reduce its harmful effects [1,4]. However, China has a long history of cannabis cultivation, and rich germplasm resources (Figure 1). Cannabis has been domesticated for stem fiber (and to a minor extent for oilseed) in Western Asia and Europe. In terms of molecular breeding technology, a mature cannabis regeneration system and space mutation germplasm resource bank have been established in China. At present, the Chinese national cannabis germplasm resource bank has more than 1500 cannabis germplasm resources, including medicinal types, seed types, and fiber types (Figure 1). This is the largest number of resources in the world and has a rich genetic diversity (Figure 1) [1,2,3,4]. Studies have shown that cannabis is mostly dioecious, with most individuals being heterozygous, and that there are many variations in the cannabis plant [1]. For another, based on actual production needs, cannabis is required to have high and stable yields; consistent plant uniformity; high quality; strong resistance; stable hereditary toxicity; strong environmental adaptability and tolerance; and adaptability to mechanization as a cash crop [1,3]. Therefore, China is one of the most important cannabis producers in the world (Figure 1) [3,4], mainly for textiles, clothing, and other woven applications [1,5,6]. In the future, China may continue to play a leading role in this field, although cannabis textiles are outdated. According to reports, 18,560 hm^2^ of land is used for cannabis cultivation in China, with a total output of 106,200 t every year, which demonstrates relatively stable growth [5,6]. Therefore, cannabis germplasm resources may be divided into wild resources, local varieties, selected varieties, strains, and genetic materials [7,8]. In addition, Chinese research on the innovative applications of cannabis genetic transformation systems and gene editing technologies is also in a leading position in the world [1,8]. On the other hand, agriculture began as early as 13,000 BC. In some places, it was the foundation of civilization. This study shows that of the thousands of plant species used by humans for various purposes, only a few dozen are essential to the progress of civilization, and cannabis is one of them [1]. In addition, while there are many different kinds of cannabis plants in China, they can be divided into three basic utilitarian categories, including three groups of cultivated plants that were selected as characteristic economic products: (1) “Wild” (weeds) plants that escaped from cultivation and grow independently in nature; (2) Fibers from the main stalks (used for textiles, ropes, and many new applications); and (3) Oilseeds (oil-rich seeds employed for human food, livestock feed, nutritional supplements, industrial oils, and occasionally as biofuels) [1,8,9] (Figure 1 and Figure 2). These are associated with the two vascular (fluid transportation) systems of plants: xylem tissue, which transports water and solutes from the roots to other parts of the plant; and phloem tissue, which transports photosynthetic metabolites from the foliage to nourish other parts of the plant [1,10,11]. Industrial cannabis refers to hemp containing less than 0.3% tetrahydrocannabinol. In China, industrial hemp is called cannabis (Figure 1). Cannabis is widely regarded as indigenous to temperate, Western, and Central Asia, but may be traced to Eastern Asia [9,10]. In addition, the data show that the majority of cultivars licensed in Western nations must by law have a THC content of less than 0.3% (dry weight) in the upper flowering portion, with regulations in some jurisdictions requiring less than 0.2% [1,11,12]. Cannabis planting mode in China could be divided into three main types: outdoor planting, indoor planting, and greenhouse planting.

Cannabinoids are terpene phenolic compounds produced during the growth and development of cannabis and are enriched in the hairs of the female flower glands of cannabis around the flowering stage of cannabis [13,14,15,16]. They are the active chemical components in the cannabis plant and the main components of cannabis that exert pharmacological activity. However, since the early 1980s, the European Union has provided large subsidies for the development of new cannabis harvesting and fiber processing technologies. Therefore, Europe (especially France) has developed non-woven applications of cannabis fiber [17,18,19,20,21]. However, due to competition with synthetic fiber and other natural fibers, the applications of cannabis fiber are very limited. Although cannabis fiber is a small crop and of relatively small importance in daily life, it has experienced a limited economic recovery based on non-traditional uses, especially in the production of a very wide range of pressed fiber and insulating products and plastics, particularly in automobiles, construction, and agriculture [20,21,22,23] (Figure 2). This study has shown that there are two main areas of use of cannabis: fiber and textiles, and general health. At present, cannabis plants and their components can only be used in the field of medicine; in traditional Chinese medicines or health foods; and as raw materials for cosmetics [20,22,24]. Industrial cannabis fiber has several unique characteristics: it is naturally antibacterial and has excellent ultraviolet-radiation-shielding, anti-static, and heat-resistant properties. Studies have identified and resolved more than 100 cannabinoids [15,16,24]. Among them, phenolic compounds are the main research objects, including cannabinol (CBN), cannabidiol (CBD), and cannabidivarin (CBDV) [17,18]. Therefore, phenolic substances represented by cannabidiol (CBD) have rich pharmacological effects [19,20]. Studies have shown that cannabidiolic acid (CBDA) is one of the most abundant cannabinoids in cannabis plants, and can be converted into CBD and CBDV by heating and decarboxylation (Figure 2).

Cannabinoids are a group of c21 terpenophenolic compounds that naturally exist in cannabis plants. They are related to the terpenes, with their ring structure derived from geranyl pyrophosphate which represents the most specific group of compounds in this plant [1,17,19]. At present, industrial cannabis and cannabis plants are mainly extracted by solvent extraction or carbon dioxide extraction. There are three types of cannabinoids in the study: synthetic, botanical, and endogenous cannabinoids. Botanical cannabinoids are plant-based substances; furthermore, their structures and functions are similar to those of general cannabinoids (Figure 3). We believe that due to the more stringent seed cultivation and growth environment requirements for the cultivation of cannabis products with high CBD (cannabidiol) content, the marginal change of cultivation purpose may also become one of the driving forces for the growth in demand for indoor and greenhouse cultivation scenarios [1,19]. However, cannabinoids are naturally produced by plants in many studies [21,22]. They are a unique active ingredient in cannabis [23,24]. There are some key factors affecting industrial cannabis breeding and industrial development [25,26]. More than 100 kinds of cannabinoids have been isolated, and the main cannabinoid components in cannabis plants include tetrahydrocannabinol (THC), cannabidiol (CBD), and cannabichromene (CBC) [27,28] (Figure 3). The research on key enzymes and their genes and inheritance methods is summarized (Figure 3). Therefore, the importance of cannabinoids is mainly reflected in their possible use in healthcare. As an alternative treatment for some health conditions, synthetic cannabinoids are also an important source of cannabinoids (Figure 3). Some individual cannabinoid molecules have been studied, and some cannabinoids have also been found to have potent medicinal properties [26]. Research has demonstrated that although cannabinoids have clinical medical value, their extraction is difficult and complicated [28,29,30]. Many studies have shown that different cannabinoids can potentially be used to treat difficult and miscellaneous diseases today [29,30,31,32]. In recent times in China, cannabis seeds (hemp seeds) have been used to treat blood problems and constipation. Cannabis is natural, environmentally friendly, and multifunctional. In particular, cannabidiol (CBD) has extremely high medical value. It is known as the second most valuable medicinal active ingredient obtained from plants in the world after artemisinin [19,26,28]. Therefore, the advantages of cannabis textiles have been confirmed by the market [28,30]. However, the potential of cannabis in the field of general health has not been fully developed. At present, tens of thousands of commercial products use cannabis extracts as raw materials in the fields of food, medicine, and medical beauty, especially for the treatment of Parkinson’s disease, Alzheimer’s disease, epilepsy, anxiety, and other diseases [24,30,32]. In addition to the roots, stems, and leaves of industrial cannabis, cannabis seeds can also be processed into feed for livestock and food for humans. At the same time, CBD extracted from industrial cannabis is also a commonly used additive in many foods. However, there is still relatively little research on the cannabinoids of cannabis. The purpose of this paper is mainly to illustrate research progress on cannabinoids in cannabis.

## 2. Main Cannabinoid Constituents

### 2.1. Cannabidiol (CBD)

Cannabidiol (CBD) is one of the most valuable active ingredients of cannabis, and has the functions of relaxing the body and mind, protecting the nerves, and improving skin inflammation, in addition to anti-oxidation effects. According to many reports, the industrial value of CBD may reach 20 billion dollars in 2024 [29,30]. However, as the main non-toxic component of cannabis, studies have shown that CBD is quite abundant in common cannabis strains, which makes it easy to convert the separation and use of cannabinoids into commercial production operations [30]. Research has confirmed that CBD has pharmacological effects, including anti-spasmodic, anti-anxiety, and anti-inflammatory effects [30,31]. Studies have shown that cannabidiol can be used in the treatment of many difficult diseases, and can also effectively eliminate the hallucinogenic effect of tetrahydrocannabinol (THC) on the human body, and is known as the “anti marijuana compound” [32,33]. In many reports, cannabidiol (CBD) is a kind of cannabinoid with anti-inflammatory properties but no psychoactive effects, and has a wide range of uses in many fields [33,34]. Although it may not make users feel excited in the traditional sense, cannabidiol has various therapeutic applications. Studies have found it is an anticonvulsant compound that can be used to treat epilepsy. For example, the FDA approved Epidiolex, a drug rich in cannabidiol, for the treatment of severe epilepsy [35,36].

Studies relating to the use of cannabidiol as an antipsychotic drug have found that it has anti-depressant and anti-anxiety effects [37,38]. It can also have a positive impact on depression. Researchers have found that cannabidiol significantly reduced subjective anxiety [37,38,39]. Cannabidiol also has the characteristics of an anti-addiction treatment, which may be the reason it is considered to be a treatment for opioid addiction [39,40]. According to many studies, cannabidiol also has a potential therapeutic effect on a wide range of neuropsychiatric diseases, and is used in the treatment of post-traumatic stress disorder (PTSD) [41,42]. In China, cannabidiol also shows medicinal ingredients as an anti-cancer drug. It has been proven that it is toxic to human breast cancer cells, could slow down the metastasis and spread of cancer cells, and can inhibit neuropathic pain. When used in combination with tetrahydrocannabinol (THC), it may alleviate the pain of patients with advanced cancer [43,44]. Scientists have synthesized a series of CBD analogues through continuous modification of the structure of CBD [37]. These analogues have different pharmacological activities and can act on different diseases [42]. Therefore, various parts of the plant were used medicinally in ancient China, including the foliage and roots. In many cases, CBD has many beneficial effects. The food and beverage industry has created the “CBD+” model. Foods and beverages with added CBD are often used to help people manage stress, relieve anxiety, relax, and improve depression and sleep.

### 2.2. Tetrahydrocannabinol (THC)

THC is used to determine the properties of cannabis. According to international regulations, the content of tetrahydrocannabinol in industrial cannabis is ≤0.3%. In order to prevent the abuse of cannabis, many countries have limited the THC content of cannabis, especially China [1,9,24]. The psychoactive component of the cannabis plant is one of the cannabinoids found in the resin glands of female cannabis stems [45,46]. Cannabis in China should not contain levels of tetrahydrocannabinol (THC) more than 0.3%. However, the anesthetic component of cannabis, tetrahydrocannabinol, was later discontinued. As a result, it has a long list of therapeutic effects, including analgesic properties, and acts as a muscle relaxant [44,47]. Studies have found that when tetrahydrocannabinol is used in combination with other cannabinoids, it can even relieve the neuropathic pain caused by multiple sclerosis [48]. However, on the other hand, tetrahydrocannabinol has also been proven to reduce intraocular pressure, which makes it a viable alternative to future anti-glaucoma drugs. Of course, tetrahydrocannabinol (THC) has been widely banned in China. Many studies have found tetrahydrocannabinol (THC) is an important anti-cancer drug [49]. It is also an antiemetic agent which can inhibit the migration of lung cancer cells (in vitro experiments) and the growth of lung adenocarcinoma cells in vivo (oral experiments in mice) in many cases [50]. However, tetrahydrocannabinol (THC) is only used in scientific research less than 0.3%, which is important for distinguishing cannabis from marijuana. China is the first country in the world to include the entire class of synthetic cannabinoids on the list of anesthetic drugs.

### 2.3. Cannabichromene (CBC)

CBC is non-psychoactive and the most common cannabinoid next to THC and CBD, and is distributed in all parts of cannabis plants according to many studies [51]. However, in some cannabis plants, the CBC content is higher than the CBD content. Studies have shown that CBC possesses anti-inflammatory, anti-tumor, antidepressant, and antifungal properties, and also promotes brain growth [37,52]. Research has shown that CBC is another minority cannabinoid that is not toxic [49]. Many studies have found CBC has analgesic, antidepressant, and anti-inflammatory effects in experimental rats. Studies have also found that CBC demonstrated anti-tumor effects against cancer cells, and a strong antibacterial effect in many cases [53].

### 2.4. Cannabigerol (CBG)

Studies have found that non-psychoactive ingredients exist in the early growth cycle of cannabis. It is therefore difficult to find a large amount of CBG in cannabis plants, which also means that its medical use can be obtained by cultivating cannabis (Figure 4). Therefore, according to many reports, CBG can be used in the treatment of psoriasis; as an antibiotic, antidepressant, and analgesic; and has anti-tumor properties [54]. Studies have shown that it can reduce inflammation, relieve pain, and even slow down the proliferation of some cancer cells, and that it has therapeutic potential [54,55]. Studies have also found that CBG is the precursor of CBGA [54,55], a precursor molecule which can develop into many different types of cannabinoids (Figure 4). Which is sometimes called the “mother of cannabinoids”. CBG is thought to elicit its therapeutic effects directly through interactions with the CB1 and CB2 cannabinoid receptors in the brain [30,55].

However, research has shown that it can still affect your mood. According to the experience of many, cannabinol can prevent panic attacks faster and more effectively than cannabidiol [56]. Studies have found that cannabinol is the precursor of CBGA, a precursor molecule, which in many cases can then develop into many different types of cannabinoids [57] (Figure 4). According to a wide body of research, CBG can prevent panic attacks faster and more effectively than CBD (Figure 4). It has a unique nerve-calming effect [45,46,47] (Figure 4). As reported, cannabis strains in China usually contain very little CBG, generally less than 1% by weight.

### 2.5. CBN (Cannabinol)

Tetrahydrocannabinolic acid (THCA) breaks down into CBN over time. Cannabinol is a medical use of cannabinoid in many cases [44]. According to studies, cannabinol can be used as an antibiotic, a potential treatment for amyotrophic lateral sclerosis, and a treatment for glaucoma, and has appetite-stimulating, analgesic, anti-asthmatic, sedative, and other effects [58]. However, this study has shown that cannabinol is a peculiar compound [35]. As reported, cannabinol can usually be found in old marijuana, because tetrahydrocannabinol (THC) can be oxidized to cannabinol in the cannabis plant [36]. Nevertheless, cannabinol is still a non-psychotropic (non-narcotic) cannabinoid [37].

Cannabinol is usually considered to be a sedative cannabinoid, which in many cases can help people to sleep. However, few studies support such claims [45]. Research shows that cannabinol can activate the CB1/CB2 receptors in vivo, which can lead to the inhibition of nerve transmission [44]. Cannabinol has anticonvulsant effects and has an effective effect on methicillin-resistant staphylococcus aureus [43]. In addition, it has anti-inflammatory properties, is considered to be a medicine for treating burns, and may have an effect on bone formation [44].

### 2.6. Cannabidivarin (CBDV)

In China, many reports have found that cannabidivarin is very similar to CBD, which is a slightly degraded version of a cannabinoid. However, this small change in molecular shape is of great significance [45]. So far, there has not been a lot of research on CBDV [59]. Research has shown that CBDV has anti-epileptic and anti-nausea effects [57]. CBDV is another non-toxic cannabinoid. According to reports, it mainly exists in “indica” Indian cannabis plants [59]. Similarly to CBD, CBDV also has anticonvulsant characteristics [43]. Research on mice has demonstrated these effects and found that CBDV (cannabidivarin) played a significant anticonvulsant role in three seizure models. The research results also point out that the findings strongly support the further clinical development of CBDV in the treatment of epilepsy [45,47]. Inspired by relevant information, the GW Pharmaceutical Company (the manufacturer of Epidiolex, an epilepsy drug rich in CBD) released a new drug patent for epilepsy based on CBDV [60].

However, CBDV has also been evaluated as a potential treatment for certain emotional and behavioral disorders, such as Reiter’s syndrome (RETT) and autism spectrum disorder (ASD) [47,51]. Some studies on autism spectrum disorder have also pointed out that the ASD mouse model demonstrates the potential therapeutic mechanism of CBDV, including its potential therapeutic effects on repetitive behavior, irritability, social interaction, quality of life, and reducing inflammation [56,57,58]. Cannabis and cannabinoids are useful in the treatment of diseases. However, the mechanisms of many treatments are still unclear [56,58].

### 2.7. Tetrahydrocannabivarin (THCV)

While some studies have shown that thypohydrocannabinol (THCV) has about 20% of the mental effect of THC, there is still no exact medical evidence [45,57,59]. In many cases, the medical value of THCV is in its anticonvulsant, weight loss, and neuroprotective effects. Therefore, cannabinoid acid is the acid form of CBD before decarboxylation [59,60,61]. Research on THCV is more limited than on other cannabinoids. As with many other members of this chemical family, THCV binds to receptors located in different organs and systems, such as the brain and the immune system in many cases [45,57,59,61]. The interaction may also have some beneficial effects on the body. THCV may help regulate blood glucose levels. One study showed that THCV significantly reduced fasting plasma glucose in type 2 diabetes patients and improved pancreatic function [54,59]. Therefore, the studies show that the THCV content of most common strains is less than 1%, which makes the cost of extracting this cannabinoid in large quantities very high [51,57,59,61]. In summary, cultivating cannabis strains with high THCV and low THC is one of the new challenges facing cannabis breeders [56,62,63,64].

### 2.8. Delta-8,Tetrahydrocannabinol Acid (*∆*8-THC)

∆8-THC is related to ∆9-THC (dronabinol), which has very little psychoactive effect on adults, and there is currently little medical research data about either of ∆8-THC and ∆9-THC [65,66]. In China, studies have found that ∆8-THC has medical value in that it can stimulate appetite and has anti-nausea properties [67]. Some sources indicate that ∆8-THC has neuroprotective and anti-anxiety properties, but more experiments are needed to verify how the cannabinoid acts in the human body. ∆8-THC has an obvious appetite-increasing effect on rodents. There is little research data on them from previous studies in China.

### 2.9. Cannabidiolic Acid (CBDA), Tetrahydrocannabinol Acid (THCA)

According to many reports, cannabinoid acid is the acid form found before the decarboxylation of CBD [53,55,59]. Similarly to THCA, it also has several unique benefits. CBDA can prevent vomiting in animal models, and its binding force is 100 times that of CBD in many cases [57]. However, CBDA may also be another kind of cannabinoid with a strong antiepileptic effect [54,55,56,57,58]. Studies have found that CBDA has a positive effect on cancer treatment when combined with other cannabinoids, and may also have an anti-anxiety effect. Therefore, CBDA and THCA are compounds found before the decarboxylation or decomposition of THC and CBD [68]. A previous study revealed that they can be found in cannabis and used as a nutritional supplement or for external use [69]. Therefore, tetrahydrocannabinol (THCA) is a nontoxic carboxylic acid (“native”) form of tetrahydrocannabinol found in cannabis plants. The decarboxylation of tetrahydrocannabinol, or heating at low temperatures, can transform the acidic tetrahydrocannabinol acid into the psychoactive tetrahydrocannabinol we are familiar with [70]. The study shows that the carboxylic acid form of cannabinoid has unique benefits, especially in terms of anti-inflammation [71]. THCA is also a neuroprotective agent with anti-inflammatory properties. In one study, researchers tested the effect of THCA on some tumor cell lines, as well as on arthritis and irritable bowel syndrome (IBS) patients [66,67,68]. In many previous studies, cannabinoid acid is the acid form before the decarboxylation of CBD. Similarly to THCA, it also has several benefits. However, the exact molecular mechanism of cannabinoid on mediate the activity remains to be elucidated [45,46,47,48]. The study also found that CBDA can prevent vomiting in animal models, and its binding force is 100 times that of CBD. However, CBDA may also be another kind of cannabinoid with strong antiepileptic effects in many cases [72]. CBDA has a positive effect on cancer treatment when combined with other cannabinoids, and may also have an anti-anxiety effect.

## 3. China’s Policy on Cannabis

As the world leader in cannabis seed production, China can produce cannabis seeds cheaply. However, the imported materials must be disinfected, which causes delays, increases costs, and reduces the quality of food [71,72,73]. The entry threshold for industrial cannabis processing in China is high, and processing licenses are scarce. China uses a licensing system for producers engaged in the processing of industrial hemp flowers and leaves [74,75]. Before the public security organ will issue a license to these producers, preparatory pre-approval is needed [75,76]. Therefore, the industrial cannabis processing license threshold is higher than that of the cultivation license. However, there are relatively few studies and reports on medicinal cannabis in China.

In China, cannabis is strictly controlled as a narcotic drug and as a class of psychotropic drugs [73,74]. At present, the relevant administrative regulation relating to the management of industrial cannabis is the Regulation on the Administration of Narcotic Drugs and Psychotropic Drugs (Figure 5) [73,77]. In March 2003, the Public Security Department of Yunnan province formulated interim regulations on the management of industrial cannabis in Yunnan province. In 2010, the Yunnan provincial government issued a document clarifying the legalization of industrial cannabis (Figure 5). On 14 September 2020, the unveiling ceremony of the Kunming international industrial cannabis industrial park was held in the Kunming area of China (Yunnan) Pilot Free Trade Zone (Kunming Economic Development Zone). The measures for the identification of industrial cannabis varieties in Heilongjiang Province and the standards for the identification of industrial cannabis varieties in Heilongjiang Province were promulgated and implemented by the agricultural commission of Heilongjiang province on 21 August 2018 (Figure 5). The purpose of these two normative documents was to maintain public safety and promote the development of the industrial cannabis industry in the province (Figure 5). In 2021, Heilongjiang agricultural department used cannabis as a rotation crop. In 2018, the Jilin provincial public security department made the “management of industrial cannabis” a separate chapter in the Jilin provincial drug control regulations (draft). In 2020, the measures for the identification of industrial hemp varieties in Jilin province clearly identified four types of variety, namely fibers, seeds, seed fibers, and cosmeceutical (Figure 5). China has a wide range of cannabis cultivation areas, forming a production layout with flowers and leaves in Yunnan province, fibers in Heilongjiang province, and grains in Shanxi province (Figure 1).

Studies have confirmed that cannabis seeds contain 25–35% oil, 20–25% protein, 20–30% carbohydrates, and about 20% other ingredients [78,79]. They are rich in more than 20 known amino acids, including nine essential amino acids [77,79]. Many reports have shown that cannabis seeds usually could be used in medicine. They are sweet in taste, flat in nature, non-toxic, and a health food derived from the same source as medicines and food. Reports have shown they have great medicinal value [77,80]. At the same time, research on industrial hemp in the food field in China has shown that industrial hemp oil contains 13–20% oleic acid, 40–60% omega-6 linoleic acid, and 15–30% omega-3 α-linolenic acid [80,81]. Studies have shown that linoleic acid and linolenic acid are polyunsaturated fatty acids that are necessary for the human body but which cannot be synthesized by themselves. They can reduce blood cholesterol and triglycerides, and have preventive and therapeutic effects on cardiovascular and cerebrovascular diseases [76,78]. Many studies have shown that industrial cannabis seeds can be processed into industrial hemp protein powder after dehulling, oil extraction and degreasing; their protein content is 55–60%, and their protein content after purification reaches 80–90% [76]. Defatted hemp seed protein powder is rich in eight kinds of essential amino acids needed by the human body and contains a variety of ingredients beneficial to the human body in the study [82]. Studies have shown that their amino acid content is more than 20%, of which essential fatty acids are more than 10% [20,80].

**Figure 5 molecules-28-03806-f005:**
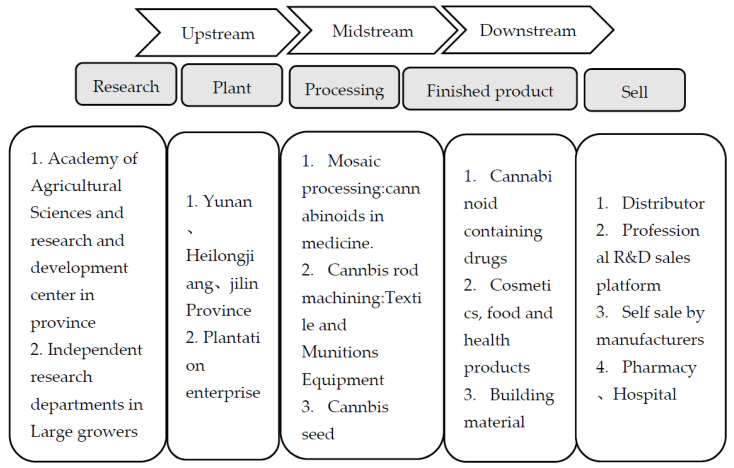
Schematic Diagram of China’s Cannabis Industry Chain [75,79,81,83,84].

At present, fiber textiles have become the main finished product of industrial cannabis, accounting for 70% [74,80]. As reported, China has become the world’s leading producer of industrial cannabis fiber [81,83]. In the domestic market, the development of the industrial cannabis industry cannot be separated from the support of national policies, but because it is a new, industry, it has mainly been promoted by the local government [76,81,83] (Figure 5). At present, there are more than ten provinces and cities in China that grow industrial cannabis [75]. Therefore, Yunnan, Heilongjiang and Jilin in China are three provinces that can legally grow industrial cannabis [75,84] (Figure 5). In recent years, with the continuous progress of extraction and purification technologies and the rise of the global industrial cannabis industry, countries have also actively promulgated a series of relevant policies to promote the legalization of cannabis and the development of related industries [74,77,78] (Figure 5). In the study, France is the largest producer of raw cannabis and industrial cannabis in the world at present, followed by China [75,81,83]. South Korea and Russia are also major industrial cannabis producers. Industrial cannabis is in short supply [75,79]. With the liberalization of industrial cannabis policies in many countries, some nations and regions are reintroducing the production of industrial cannabis [79,82,85] (Figure 5). There are currently 31 countries legalizing industrial cannabis cultivation [75,83]. In recent years, many countries have gradually lifted their bans on the medical use of cannabis, and many products have appeared around the world [86,87]. However, its status in China is still embarrassing.

## 4. Conclusions

Cannabis contains many active compounds, of which cannabinoids are the main active components. Cannabinoids have similar molecular structures, but their real difference centers on their psychoactive properties. Unfortunately, cannabidiol and cannabinol were the only components investigated that were not psychoactive. Therefore, the use of THC, CBN, and CBC is limited to possible therapeutic purposes. There are more than 400 compounds in cannabis plants, and while individual compounds have not been studied, these are enough for us to examine the medical performance of cannabis plants. However, most of the industrial cannabis planted in China is still used for fiber, with little or no CBD. Therefore, China needs more seeds with high CBD content than other countries. Cannabinol is the main non-psychoactive component of cannabis, and its anti-tumor activity in vitro and in vivo is considered to make it an anti-tumor drug. As the applications of cannabinol in the fields of medicine and healthcare are gradually being recognized by the market, industrial cannabis and related industries have broad development space.

## Figures and Tables

**Figure 1 molecules-28-03806-f001:**
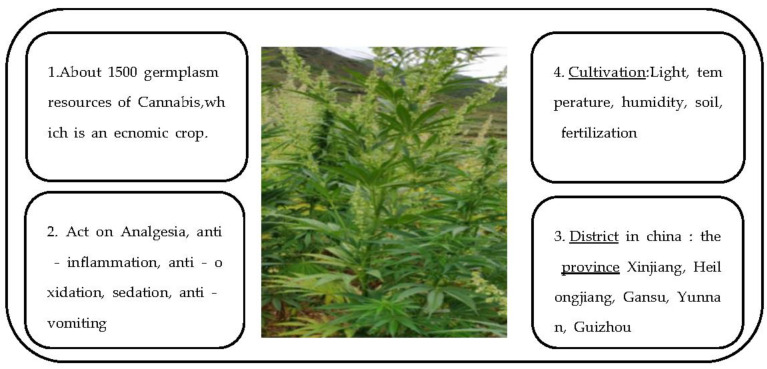
Distribution and application of cannabis in China [1,4].

**Figure 2 molecules-28-03806-f002:**
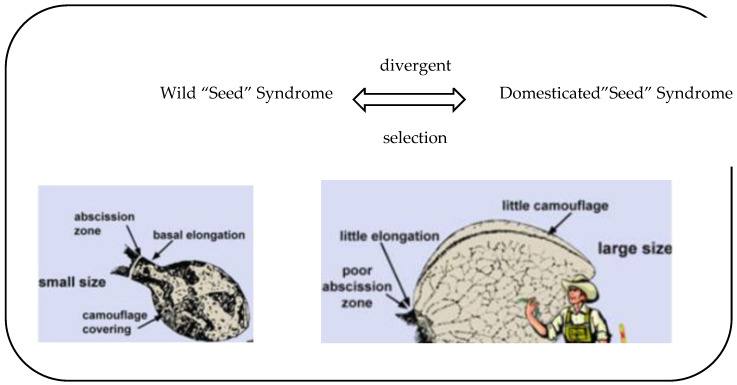
Divergent selection for adaptive achene (“seed”) characteristics between domesticated and wild plants of cannabis [1].

**Figure 3 molecules-28-03806-f003:**
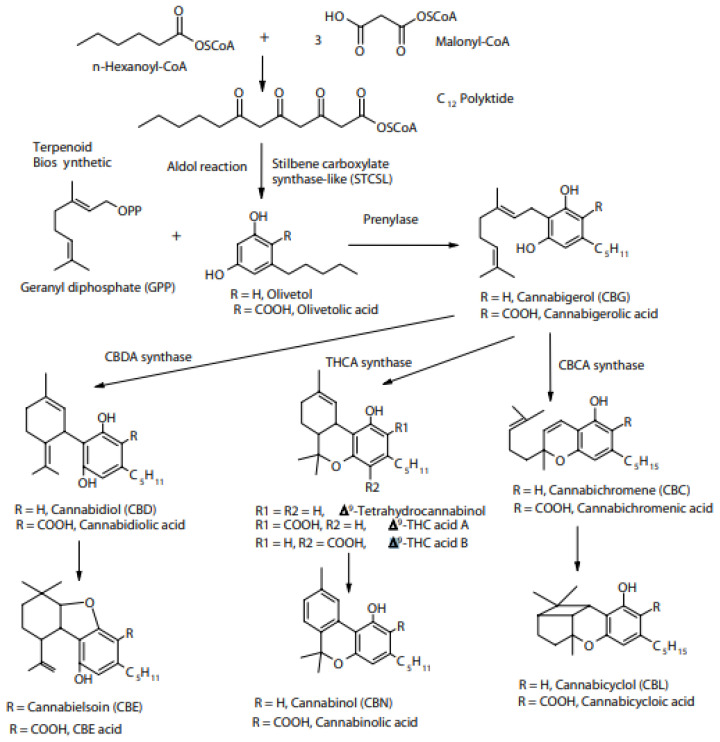
Biosynthesis of cannabinoids in cannabis sativa. The cannabinoid acids are formed in the biosynthetic process, while corresponding decarboxylation products are formed later by decomposition, e.g., under the influence of heat [8].

**Figure 4 molecules-28-03806-f004:**
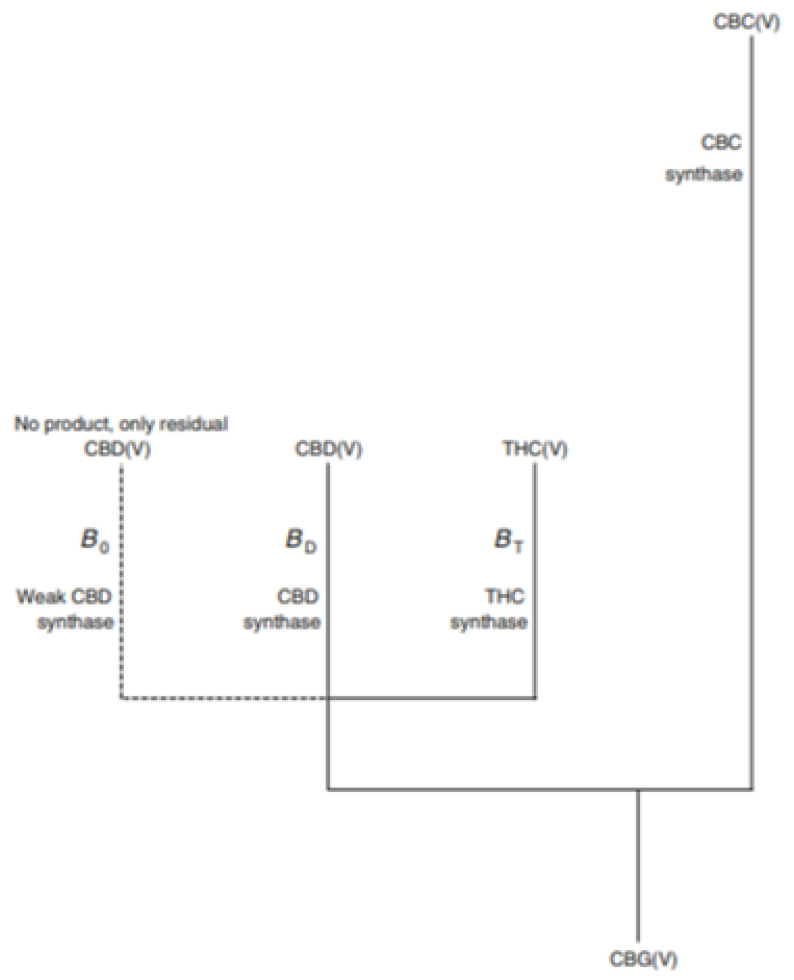
Updated model for the regulation of the different conversions of CBG(V) by the independent loci B and C. Locus B has two common alleles, BD and BT, responsible for the conversion of CBG(V) into CBD(V) and THC(V), respectively [20].

## Data Availability

Data available in a publicly accessible repository.

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
