# Peer review of "Research Progress on Cannabinoids in Cannabis (Cannabis sativa L.) in China"

_molecules, 2023, doi:10.3390/molecules28093806_

Round 1

Reviewer 1 Report (Previous Reviewer 2)

In general, the document has improved the writing and handling of the subject regarding the use of cannabis in China, it seems to me that the modifications made up to now have favored the improvement of the quality of the manuscript.

However, reviewing the document again I found some details that can be improved, basically on the writing.

In this sense, the authors repeat the scientific name of the plant several times, however, they do it wrongly, since all scientific names must be written in italics. In the same way, the name of the species, in this case Cannabis, always goes with the first letter in capital letter. Also, it is necessary that the authors consider that once the scientific name is mentioned in the document, the other times it is written, it must be abbreviated as follows: place only the first letter of the genus followed by a period (C.) and later the full species name, all in italics.

Also, I still think that the title of figure 1 does not match what is in said image, therefore, I think it would be better to change it to "scheme", since it is not an image as such.

Author Response

Open Review

English language and style

( ) English very difficult to understand/incomprehensible
( ) Extensive editing of English language and style required
( ) Moderate English changes required
( ) English language and style are fine/minor spell check required
(x) I don't feel qualified to judge about the English language and style

Is the work a significant contribution to the field?

Is the work well organized and comprehensively described?

Is the work scientifically sound and not misleading?

Are there appropriate and adequate references to related and previous work?

Is the English used correct and readable?

Comments and Suggestions for Authors

I believe that the pertinent, and requested, corrections to the manuscript have been made. For my part I think that the way it is now it can be published.

Submission Date

01 February 2023

Date of this review

10 Feb 2023 15:43:33

Author’s Reply to the Review Report:

Institute of Bast Fiber Crops, Chinese Academy of Agricultural Sciences has successfully obtained gene-edited plants and stable transgenic plants with kanamycin resistance for the first time in industrial Cannabis. At the same time, we established Chuxiong Southwest Industrial Cannabis Research Center and North Industrial Cannabis Research Center. Therefore, our research on industrial Cannabis will be the focus of our future research. Thank you for very much. We will contribute to the development of industrial Cannabis. Of course, the English level of the paper has also been revised for the publish, thank you again.

Reviewer 2 Report (Previous Reviewer 3)

Review comments to the author

Title: ''Research progress of cannabinoids in Cannabis (Cannabis sativa L.) of China''.

Manuscript ID: molecules-2224323.

The work has been carried out correctly and there is nothing wrong with the results.

Recommendation:

Publish without any further revision.

Author Response

Open Review

English language and style

( ) English very difficult to understand/incomprehensible
( ) Extensive editing of English language and style required
( ) Moderate English changes required
(x) English language and style are fine/minor spell check required
( ) I don't feel qualified to judge about the English language and style

Is the work a significant contribution to the field?

Is the work well organized and comprehensively described?

Is the work scientifically sound and not misleading?

Are there appropriate and adequate references to related and previous work?

Is the English used correct and readable?

Comments and Suggestions for Authors

Review comments to the author

Title: ''Research progress of cannabinoids in Cannabis (Cannabis sativa L.) of China''.

Manuscript ID: molecules-2224323.

The work has been carried out correctly and there is nothing wrong with the results.

Recommendation:

Publish without any further revision.

Submission Date

01 February 2023

Date of this review

14 Feb 2023 09:26:30

Author’s Reply to the Review: Cannabis contains more than 500 chemical components, making it one of the most complex plants. These ingredients are divided into cannabinoids and non-cannabinoids. There are more than 100 kinds of cannabinoids. Humans have cultivated cannabis for about 5,000 years. China is the origin of cannabis. There are a large number of wild germplasm resources in the wild. In addition to the germplasm resources imported from abroad, more than 1,500 copies of wild cannabis have been preserved in the national cannabis p germplasm resource bank of the Cannabis Institute. Seeds, which are the main material for cannabis breeding and scientific research. However, there are relatively few studies on cannabinoids, especially in China, and there are few reports 。Thank you very much。

Reviewer 3 Report (New Reviewer)

Manuscript ID molecules 2224323

Author Response

Open Review

English language and style

( ) English very difficult to understand/incomprehensible
(x) Extensive editing of English language and style required
( ) Moderate English changes required
( ) English language and style are fine/minor spell check required
( ) I don't feel qualified to judge about the English language and style

Is the work a significant contribution to the field?

Is the work well organized and comprehensively described?

Is the work scientifically sound and not misleading?

Are there appropriate and adequate references to related and previous work?

Is the English used correct and readable?

Comments and Suggestions for Authors

Manuscript ID molecules 2224323

Submission Date

01 February 2023

Date of this review

16 Feb 2023 18:55:32

Manuscript ID molecules-2224323 The author aims to present the botanical characteristics and research progress on the chemical composition of cannabis sativa cultivated in China, as highlighted in the title. The novelty and originality of this manuscript is poor and the topics discussed overlap with other reviews on the same subject already in the literature. The manuscript can be accepted with major revisions and needs improvements focusing on food use in China or other developments. General comments This is a review. The author describes natural products already isolated from cannabis varieties cultivated in other geographical areas and reports biological activities known from the literature. There is no indication of new compounds or biological activities or uses other than those already widely reported in previous reviews.

Author’s Reply to the Review: Cannabinoids are the main active ingredients of cannabis, which have analgesic, anti-inflammatory, antioxidant, sedative, and anti-emetic effects. According to the structural characteristics of the compounds, the main cannabinoid components in cannabis are summarized in this paper. Our institute has always been in the leading position in China, but we have not systematically reported the research progress of cannabinoids in cannabis. It is of reference significance to speed up the research on cannabinoids and cannabis breeding in china.

 I believe that the manuscript doesn’t meet the aim and scope of Molecules/Food chemistry in its current form, This manuscript was submitted to the section Food chemistry but does not deal with the use of cannabis as a foodstuff in China and there is no evaluation of functional components and their multifunctional applications. The discussion and Conclusion show that Chinese cannabis is only used as a textile fibre.

Author’s Reply to the Review: Cannabinoids are the main components of cannabis to exert pharmacological activity. More than 100 cannabinoid compounds have been isolated and identified, among which tetrahydrocannabinol (THC) and cannabidiol (CBD) are the main ones. But there are not many reports about cannabinoids at home and abroad. China is the first country to regulate synthetic cannabinoids. This paper summarizes the cannabinoid components in cannabis and their main research progress.

The text on page 10 is not related to the scientific content.

Author’s Reply to the Review: I have made serious revisions. The main purpose of page 10 is to introduce the development of industrial hemp in China. As you can see, the application of industrial hemp in the world is restricted, especially in China, so we want to To expand the development of industrial hemp, it is necessary to quickly understand the industrial hemp development policies of each country. The main purpose of this article is to introduce the research progress of cannabinoids in industrial hemp in China, so the main purpose of Page 10 is to explain the development of industrial hemp in China. Development, I hope my answer can get your satisfaction.

Many references cited in the main text do not correspond to the numeration in the reference list. From reference 35 there is no longer a match.

Author’s Reply to the Review: Thank you for your comments, I have made serious revisions, and for such errors, I have completely modified them according to your comments.

Please check ref 24 line 158, ref. 35,36,60 for Epidiolex, ref 50 line 218, line 271 ref 43, line 307 ref 54, 59 etc…

Author’s Reply to the Review: I have made serious revisions, thank you for your comments, I have made adjustments where there are discrepancies in the literature, thank you again, you can refer to the revised draft for details. ref 20 line 158 .

 Some specific comments are listed below:

Author’s Reply to the Review: I have made serious revisions and fully adopted your comments. You can refer to the revised draft for details. I hope that my revisions will satisfy you.

Line 91 need a reference (decarbossilation)

Author’s Reply to the Review: I have made serious revisions. For details, you can see the revised draft. I have fully adopted your comments. I hope that my revisions will satisfy you. Thank you again

Line 158 “There is still relatively little research on cannabinoids”…. this is not true because in the last five years many reviews and papers have been reported on this subject. Line 176 : CBD cannabidiol

Author’s Reply to the Review: I have made serious revisions and fully adopted your comments. Please refer to the revised draft for details. I hope my revisions will satisfy you. In the past five years, the international literature has mainly focused on CBD, and there are very few articles that comprehensively and systematically introduce cannabinoids, especially in China. I hope my answer can get your affirmation and support

Line 228 CBC is cannabicromene

Author’s Reply to the Review: I have made serious revisions. For details, you can see the revised draft. I have fully adopted your comments. I hope that my revisions will satisfy you. Thank you again, I hope that your work will make the research of cannabinoids to a higher level.

Line 279 CVBD is cannabidivarin

Author’s Reply to the Review: I have made serious revisions. For details, you can see the revised draft. I have fully adopted your comments. I hope that my revisions will satisfy you. Thank you again, Your help and support will definitely promote the development of industrial hemp in China. Cannabidivarin (CBDV) is a phytocannabinoid naturally occurring in the cannabis plant.

 Line 414 …Conclusion…. “Cannabinoids are found in other plants, such as azalea, licorice and liver grass, and earlier echinacea”…. this part must be placed in the Introduction and not in the Conclusion and must be appropriately referenced.

Author’s Reply to the Review: I have made serious revisions. For details, you can see the revised draft. I have fully adopted your comments. I hope that my revisions will satisfy you. Thank you again, I edited it exactly as per your comments. I hope that the revised draft can be satisfactory, thank you again

Reviewer 4 Report (New Reviewer)

In this review, author aimed to provide an overview of research progress of cannabinoids in cannabis sativa. This review focus on some chemical and biological properties of cannabinoids and their application in the field of medicine and health care.

Overall, this is a hot and interesting topic. However, the manuscript cannot be suitable for publication in its present form. Furthermore, the author uploaded a version of its manuscript using a track changes mode in MS Word instead of uploading a “clean” copy of the manuscript.

Major comments:

The manuscript is a succession of bibliographical references with a sore lack of harmony and connectivity. The manuscript needs a deep revision and an extensive English editing. 

The Abstract needs to be rephrased and reorganized. Some comments have been made in attached file.

Introduction section is very difficult to understand. It needs a deep revision and reorganization. I feel that we are "jumping" from general to specific information without any connectivity and in some cases, we are “travelling” from general to specific to general (again) to specific information without any logical sequence. I have included some suggestions to reorganize this section in attached file. 

The meaning of some sentences is not clear. Please see attached file and amend when indicated. 

Minor comments:

Please add a space when citing references in main text. Apply for the entire manuscript.

Please delete repetitions when indicated in the attached file.

Author Response

Open Review

English language and style

(x) English very difficult to understand/incomprehensible
( ) Extensive editing of English language and style required
( ) Moderate English changes required
( ) English language and style are fine/minor spell check required
( ) I don't feel qualified to judge about the English language and style

Is the work a significant contribution to the field?

Is the work well organized and comprehensively described?

Is the work scientifically sound and not misleading?

Are there appropriate and adequate references to related and previous work?

Is the English used correct and readable?

Comments and Suggestions for Authors

In this review, author aimed to provide an overview of research progress of cannabinoids in cannabis sativa. This review focus on some chemical and biological properties of cannabinoids and their application in the field of medicine and health care.

Overall, this is a hot and interesting topic. However, the manuscript cannot be suitable for publication in its present form. Furthermore, the author uploaded a version of its manuscript using a track changes mode in MS Word instead of uploading a “clean” copy of the manuscript.

Major comments:

The manuscript is a succession of bibliographical references with a sore lack of harmony and connectivity. The manuscript needs a deep revision and an extensive English editing. 

The Abstract needs to be rephrased and reorganized. Some comments have been made in attached file.

Introduction section is very difficult to understand. It needs a deep revision and reorganization. I feel that we are "jumping" from general to specific information without any connectivity and in some cases, we are “travelling” from general to specific to general (again) to specific information without any logical sequence. I have included some suggestions to reorganize this section in attached file. 

The meaning of some sentences is not clear. Please see attached file and amend when indicated. 

Minor comments:

Please add a space when citing references in main text. Apply for the entire manuscript.

Please delete repetitions when indicated in the attached file.

Submission Date

01 February 2023

Date of this review

18 Feb 2023 19:54:01

Author‘s Reply to the Review :’

  1. The manuscript is a succession of bibliographical references with a sore lack of harmony and connectivity. The manuscript needs a deep revision and an extensive English editing. 

Author‘s Reply to the Review :’ I have made serious revisions, and your opinion is very important, but research on cannabinoids is still relatively lacking in China, and the main application is CBD, so our main idea is to review the lack of systematic reports and research on cannabinoids in cannabis The role of the compound in the actual production, further confirming the role of cannabinoids in cannabis. The Introduction and Abstract sections have been revised, and grammatical issues in the article have also been corrected.

  1. The Abstract needs to be rephrased and reorganized. Some comments have been made in attached file.

Author‘s Reply to the Review :’ I have made serious revisions, thank you for your comments, specifically modifying the content and grammar of the abstract. At the same time, it has been revised strictly according to your comments.

  1. Introduction section is very difficult to understand. It needs a deep revision and reorganization. I feel that we are "jumping" from general to specific information without any connectivity and in some cases, we are “travelling” from general to specific to general (again) to specific information without any logical sequence. I have included some suggestions to reorganize this section in attached file. 

Author‘s Reply to the Review :’ I have made serious revisions. The main content of the introduction part has been carefully revised according to your comments. At the same time, some mistakes have been revised strictly according to your opinions. Thank you again

  1. The meaning of some sentences is not clear. Please see attached file and amend when indicated. 

Author‘s Reply to the Review :’ I have made serious revisions. Thank you for your comments. The comments are very specific and detailed. After reading them, I don’t know how to express my thanks. I have made detailed revisions according to your opinions. At the same time, your opinions are in I have absorbed and adopted all my articles

  1. Please add a space when citing references in main text. Apply for the entire manuscript.

Please delete repetitions when indicated in the attached file.

Author‘s Reply to the Review :’ I have made very serious revisions, and I have made careful revisions according to your comments. See the revised draft for details. At the same time, thank you for your opinions, which are very specific and moving. All opinions are revised according to the revisions. I hope my modification can satisfy you.

Reviewer 5 Report (New Reviewer)

Review for the article entitled

Research progress of cannabinoids in cannabis

( 2 cannabis sativa L.) of China

Author: Sun Xiang-ping

The manuscript addresses the basic details of Cannabis sativa L. in depth. This paper reports on cannabinoid compounds produced during the growth and development of cannabis and their detailed uses in different sectors.

Major Review:

1.     The abstract was vague and should reflect the content of the paper.

2.     In Fig.1, the picture can be modified such that it clearly shows the distribution and application of cannabis. 

3.     The paper is generally well written but the literature regarding the onsearch in cannabis and the compounds derived from the plant could be discussed more.

4.     This paper clearly discusses the main cannabinoid constituents and their applications and research.

5.     It is suggested to abbreviate the repetitively used word for better appreciation in review papers. The quality of the figure resolution can still be better.

6.     I recommend the author provide details about the cannabis policies in china.

7.     The discussion may include how the other medicinal plant product policies vary with cannabis products.

8.     Also, the author can recommend various ways or suggestions to improve the awareness of the government of china.

9.     Especially, I also suggested citing more relevant and recent literature on the domestic and foreign rescue progress in china.

10.   In my view, I recommend this article after major for further consideration for publication.

Author Response

Open Review

English language and style

( ) English very difficult to understand/incomprehensible
( ) Extensive editing of English language and style required
(x) Moderate English changes required
( ) English language and style are fine/minor spell check required
( ) I don't feel qualified to judge about the English language and style

Is the work a significant contribution to the field?

Is the work well organized and comprehensively described?

Is the work scientifically sound and not misleading?

Are there appropriate and adequate references to related and previous work?

Is the English used correct and readable?

Comments and Suggestions for Authors

Review for the article entitled

Research progress of cannabinoids in cannabis

( 2 cannabis sativa L.) of China

Author: Sun Xiang-ping

The manuscript addresses the basic details of Cannabis sativa L. in depth. This paper reports on cannabinoid compounds produced during the growth and development of cannabis and their detailed uses in different sectors.

Major Review:

  1. The abstract was vague and should reflect the content of the paper.
  2. In Fig.1, the picture can be modified such that it clearly shows the distribution and application of cannabis. 
  3. The paper is generally well written but the literature regarding the onsearch in cannabis and the compounds derived from the plant could be discussed more.
  4. This paper clearly discusses the main cannabinoid constituents and their applications and research.
  5. It is suggested to abbreviate the repetitively used word for better appreciation in review papers. The quality of the figure resolution can still be better.
  6. I recommend the author provide details about the cannabis policies in china.
  7. The discussion may include how the other medicinal plant product policies vary with cannabis products.
  8. Also, the author can recommend various ways or suggestions to improve the awareness of the government of china.
  9. Especially, I also suggested citing more relevant and recent literature on the domestic and foreign rescue progress in china.
  10. In my view, I recommend this article after major for further consideration for publication.

Submission Date

01 February 2023

Date of this review

17 Feb 2023 09:25:35

  1. The abstract was vague and should reflect the content of the paper.

Author’s Reply to the Review: Thank you, I have made serious revisions, and my abstract part mainly summarizes the main content of the paper. I hope my opinions can be satisfied with you. See the revised draft for details

  1. In Fig.1, the picture can be modified such that it clearly shows the distribution and application of cannabis. 

Author’s Reply to the Review: Thank you for your opinion. I have made very detailed revisions. I hope that some revisions are not good. See the revised draft for details

  1. The paper is generally well written but the literature regarding the onsearch in cannabis and the compounds derived from the plant could be discussed more.

Author’s Reply to the Review: Thank you for your opinion, the extraction of cannabis compounds has common plant extraction and green extraction in chemistry

  1. This paper clearly discusses the main cannabinoid constituents and their applications and research.

Author’s Reply to the Review: Therefore, there are many articles and reports on the extraction of compounds, so the main function and formation mechanism of the compounds are mainly explained in the review.

  1. It is suggested to abbreviate the repetitively used word for better appreciation in review papers. The quality of the figure resolution can still be better.

Author’s Reply to the Review: Thank you for your comments, I have made detailed revisions, your comments are very pertinent, I hope my answer can satisfy you.

  1. I recommend the author provide details about the cannabis policies in china.

Author’s Reply to the Review: Thank you for your comments. I have made detailed revisions. China’s industrial cannabis policy has been introduced in detail in the article. At present, the introduction of cannabis in China is the content of the article. Only the use of industrial cannabis is prohibited in cosmetics. Cannabis is not included, and specific industrial cannabis policy explanations are not.

  1. The discussion may include how the other medicinal plant product policies vary with cannabis products.

Author’s Reply to the Review: At present, the research on industrial hemp mainly focuses on the application of CBD, including daily life and some industrial fields. There are still relatively few medicinal industrial hemp at present. I have made detailed revisions according to your opinions. I hope the answer can satisfy you.

  1. Also, the author can recommend various ways or suggestions to improve the awareness of the government of china.

Author’s Reply to the Review: Thank you for your opinion, we have been promoting the development of industrial cannabis from different angles

  1. Especially, I also suggested citing more relevant and recent literature on the domestic and foreign rescue progress in china.

Author’s Reply to the Review: Thank you for your opinion, I have fully adopted your opinion, I hope my modification can get your satisfaction

  1. In my view, I recommend this article after major for further consideration for publication.

Author’s Reply to the Review: Thank you for your comments. I have made serious revisions according to your opinions. I hope my revisions can promote the development of our industrial cannabis. I hope that our industrial cannabis research institute can promote the development of industrial cannabis from different angles and methods. Thanks again. We currently have an industrial cannabis research center and an industrial cannabis research institute, and at the same time have done a lot of work in the industrial cannabis association

Round 2

Reviewer 3 Report (New Reviewer)

Although the manuscript has been revised, no substantial changes have been made to the previous version. The manuscript does not meet the aim and scope of Molecules: Food Chemistry section,  in its current form, because it does not deal with the use of cannabis as a food in China and does not contain an evaluation of functional components and their multifunctional applications. In general, it is a review of the history of cannabis in China and its uses as a textile material, with speculation on other uses in the near future. The review deals generally of the phytochemical compounds present in cannabis, but does not report appropriate studies in China and does not take into account recent publications on the subject with isolation of minor components. (see for example: J. Nat. Prod. 2022, 85, 1089; Plants 2022, 11,1671; Cannabis Cannabinoid Res. 2020, 6, 288; Food Funct. 2018, 9, 6608….etc)

The manuscript is rejected.

Author Response

Open Review

Quality of English Language

( ) English very difficult to understand/incomprehensible
( ) Extensive editing of English language and style required
(x) Moderate English changes required
( ) English language and style are fine/minor spell check required
( ) I am not qualified to assess the quality of English in this paper

Is the work a significant contribution to the field?

Is the work well organized and comprehensively described?

Is the work scientifically sound and not misleading?

Are there appropriate and adequate references to related and previous work?

Is the English used correct and readable?

Comments and Suggestions for Authors

Although the manuscript has been revised, no substantial changes have been made to the previous version. The manuscript does not meet the aim and scope of Molecules: Food Chemistry section,  in its current form, because it does not deal with the use of cannabis as a food in China and does not contain an evaluation of functional components and their multifunctional applications. In general, it is a review of the history of cannabis in China and its uses as a textile material, with speculation on other uses in the near future. The review deals generally of the phytochemical compounds present in cannabis, but does not report appropriate studies in China and does not take into account recent publications on the subject with isolation of minor components. (see for example: J. Nat. Prod. 2022, 85, 1089; Plants 2022, 11,1671; Cannabis Cannabinoid Res. 2020, 6, 288; Food Funct. 2018, 9, 6608….etc)

The manuscript is rejected.

Submission Date

01 February 2023

Date of this review

03 Mar 2023 11:42:35

Author’s Reply to the Review:

At present, the allowed applications of cannabis plants and their components in the fields of medicine, daily use and food only include: cannabis seeds can be used as traditional Chinese medicine or health food; As a raw material for cosmetics. Studies have confirmed that hemp seeds contain 25-35% oil, 20-25% protein, 20-30% carbohydrates, and about 20% other ingredients. They are rich in more than 20 known amino acids, including 9 essential amino acids.

Many reports have shown that cannabis seed is also called cannabis seed in medicine. It is sweet in taste, flat in nature, non-toxic, and a health food with the same source of medicine and food. deficiency and other effects, great medicinal value. At the same time in China, research on industrial cannabis in the food field shows that industrial hemp oil contains 13-20% oleic acid, 40-60% omega-6 linoleic acid, and 15-30% omega-3 α-linolenic acid. Linoleic acid and linolenic acid are polyunsaturated fatty acids that are necessary for the human body but cannot be synthesized by themselves. They can reduce blood cholesterol and triglycerides, and have preventive and therapeutic effects on cardiovascular and cerebrovascular diseases.

Many studies have shown that industrial cannabis seeds can be processed into industrial cannabis protein powder after dehulling, oil extraction and degreasing, the protein content is 55-60%, and the protein content after purification reaches 80-90%. Defatted cannabis seed protein powder is rich in 8 kinds of essential amino acids needed by the human body and a variety of ingredients beneficial to the human body. The amino acid content is more than 20%, of which the essential fatty acids are more than 10%. The article mainly introduces the medicinal value of Cannabis, which is relatively rare in the food field, but the article has already added the content that was closed first. And carefully modify it on the line, Thank you for your suggestion.

Reviewer 4 Report (New Reviewer)

I went through the revised version of the manuscript. Author made substantial changes and manuscript has been highly improved and can be accepted in its present form.

Author Response

Open Review

Quality of English Language

( ) English very difficult to understand/incomprehensible
( ) Extensive editing of English language and style required
( ) Moderate English changes required
(x) English language and style are fine/minor spell check required
( ) I am not qualified to assess the quality of English in this paper

Is the work a significant contribution to the field?

Is the work well organized and comprehensively described?

Is the work scientifically sound and not misleading?

Are there appropriate and adequate references to related and previous work?

Is the English used correct and readable?

Comments and Suggestions for Authors

I went through the revised version of the manuscript. Author made substantial changes and manuscript has been highly improved and can be accepted in its present form.

Submission Date

01 February 2023

Date of this review

01 Mar 2023 18:23:42

Author’s Reply to the Review: For a long time, affected by policy and industry orientation, China's industrial Cannabis breeding has mainly been fiber and oil varieties. At present, the number of domestic medicinal industrial Cannabis promotion varieties is small, and the content of cannabidiol is low, only about 1%, which is significantly different from the international level. In recent years, with the discovery of the medicinal value of cannabidiol, a global boom in industrial Cannabis has emerged. Researchers at the Cannabis Research Institute of the Chinese Academy of Agricultural Sciences have made full use of the advantages of germplasm resources to speed up the pace of variety breeding and seize the medicinal industrial Cannabis varieties The commanding heights of breeding. Thank you for your opinion, thanks again for your opinion.

For the revision of the article, I added some applications of cannabis in the food field, and thank you again for your comments

Reviewer 5 Report (New Reviewer)

The manuscript authored by Sun Xiang-ping, addresses the document of the basic details of Cannabis sativa L. in depth. This paper reports on cannabinoid compounds produced during the growth and development of cannabis and their detailed uses in different sectors. The authors explains that 31 countries legalizing industrial cannabis cultivation. Cannabis is cultivated (or wild) all over the world, and is mainly distributed in Asia, especially in China. Cannabis fiber has been used for textile production for 6000 years in China.  The author concludes that cannabinol is the main non psychoactive component of cannabis, and its anti-tumor activity in vitro and in vivo is considered as an anti-tumor drug. With the application of cannabinol in the field of medicine and health care gradually recognized by the market, industrial cannabis and related industries have broad development space. This paper reviews the domestic and foreign rescue progress of cannabinoids in cannabis sativa, expecting to provide help for the research and development of cannabis. After the revision, the manuscript has been revised according to the suggestion. In my view, I recommend this article for publication.

Author Response

Open Review

Quality of English Language

( ) English very difficult to understand/incomprehensible
(x) Extensive editing of English language and style required
( ) Moderate English changes required
( ) English language and style are fine/minor spell check required
( ) I am not qualified to assess the quality of English in this paper

Is the work a significant contribution to the field?

Is the work well organized and comprehensively described?

Is the work scientifically sound and not misleading?

Are there appropriate and adequate references to related and previous work?

Is the English used correct and readable?

Comments and Suggestions for Authors

The manuscript authored by Sun Xiang-ping, addresses the document of the basic details of Cannabis sativa L. in depth. This paper reports on cannabinoid compounds produced during the growth and development of cannabis and their detailed uses in different sectors. The authors explains that 31 countries legalizing industrial cannabis cultivation. Cannabis is cultivated (or wild) all over the world, and is mainly distributed in Asia, especially in China. Cannabis fiber has been used for textile production for 6000 years in China.  The author concludes that cannabinol is the main non psychoactive component of cannabis, and its anti-tumor activity in vitro and in vivo is considered as an anti-tumor drug. With the application of cannabinol in the field of medicine and health care gradually recognized by the market, industrial cannabis and related industries have broad development space. This paper reviews the domestic and foreign rescue progress of cannabinoids in cannabis sativa, expecting to provide help for the research and development of cannabis. After the revision, the manuscript has been revised according to the suggestion. In my view, I recommend this article for publication.

Submission Date

01 February 2023

Date of this review

03 Mar 2023 07:37:46

Author’s Reply to the Review: China is a traditional planting country of industrial Cannabis, with a wide area and various varieties of Cannabis. This paper analyzes the research progress of industrial Cannabis cannabinoids in my country, in order to provide new ideas for the industrialization and cultivation of industrial Cannabis. The project of the Industrial Cannabis R&D Center of the Cannabis Research Institute of the Chinese Academy of Agricultural Sciences has carried out breeding research in Chuxiong City. The Cannabis Research Institute of the Chinese Academy of Agricultural Sciences has Qingma, Hanma, Jinma, Yunma series mainly for fiber, Longma, Fenma series mainly for seeds, and Chinese Hanma series mainly for medicinal purposes. Thanks.

For the revision of the article, I added some applications of cannabis in the food field, and thank you again for your comments

This manuscript is a resubmission of an earlier submission. The following is a list of the peer review reports and author responses from that submission.

Round 1

Reviewer 1 Report

The manuscript still has some major issues related to writing (typing errors), seems that the author did not aware with that problem and did not take any serious effort to fix such problem. For instance

L9 (uesd),

L21 and 22: (Cannabis sativa L) is repeated twice in the bracket.

L25 (Asia , Europe)

L27: than1,500

L27: 252 countries?? There are less than 200 countries exist in this world. 

L39: the references are not accurate! The ref given are not about law/ regulation. 

L48: etc[17-18].Phenolic

L58: (Figure 3) . (Also L64 and L65)

L65: Therefor, 

The introduction is poorly execute, does not contain the essential reason, state of the art of the review, and purpose. 

I conclude that the author did not make serious improvement on the manuscript since many basic problems occur thorough the manuscript. 

Author Response

Open Review

English language and style

( ) English very difficult to understand/incomprehensible
( ) Extensive editing of English language and style required
( ) Moderate English changes required
( ) English language and style are fine/minor spell check required
(x) I don't feel qualified to judge about the English language and style

Is the work a significant contribution to the field?

Is the work well organized and comprehensively described?

Is the work scientifically sound and not misleading?

Are there appropriate and adequate references to related and previous work?

Is the English used correct and readable?

Comments and Suggestions for Authors

The manuscript still has some major issues related to writing (typing errors), seems that the author did not aware with that problem and did not take any serious effort to fix such problem.

Reply for the review: As one of the earliest cultivated crops, cannabis has been cultivated for more than 8000 years. Modern scientific research shows that cannabis has the functions of moisture absorption, air permeability, mildew prevention, radiation prevention, UV resistance, absorption of toxic and harmful gases, and beneficial to human metabolism. With the improvement of people's living quality and consumption level, as well as the increasing emphasis on their own health and living environment, cannabis products are more and more popular, thus promoting the development and expansion of cannabis crop industry . With the continuous development and expansion of the industrial cannabis market, the legalization of the industrial cannabis industry is also being promoted. Relevant departments have issued laws and regulations to further standardize the cultivation and processing of industrial cannabis. The major cannabis producing areas in Heilongjiang, Yunnan, Jilin, Guangxi, Shanxi and other provinces of China, in combination with their own geographical and ecological conditions, excellent varieties, high yield and efficient cultivation technology, advanced breeding methods and by-product research and development, vigorously develop the industrial cannabis industry and improve economic benefits. However, under the policy environment of strict supervision, China's industrial cannabis production efficiency is generally low, mainly for export, and there are many restrictions.

For instance

L9 (uesd),

Reply for the review: is widely uesd at home and abroad

 is widely utilized at home and abroad

L21 and 22: (Cannabis sativa L) is repeated twice in the bracket.

Reply for the review: and utilization value[1]. Cannabis(Cannabis sativa) has

and utilization value[1]. For instance, It has

L25 (Asia, Europe)

Reply for the review: It is cultivated (or wild) all over the world, which is mainly distributed in Asia, Europe and America.

     It is cultivated (or wild) all over the world, which is mainly distributed in Asia, especially in china.

Europe, China, South Korea and Russia are the major cannabis production areas in the world, of which China has the largest planting area, accounting for about half of the world.

L27: than1,500

Reply for the review: than1,500

 than one thousand and five hundred

L27: 252 countries?? There are less than 200 countries exist in this world. 

Reply for the review: 252 countries?? There are less than 200 countries exist in this world. 

Chinese national cannabis germplasm resource bank has more than 1,300 cannabis germplasm resources, including medicinal types, seed types, and fiber types. Which is the largest number of resources in the world and has rich genetic diversity.

L39: the references are not accurate! The ref given are not about law/ regulation. 

Reply for the review: the references are not accurate! The ref given are not about law/ regulation. The data shows that the majority of cultivars licensed in Western nations by law must have a content of less than 0.3% THC (dry weight) in the upper, flowering portion, and in some jurisdictions regulations require less than 0.2%[11-12].

The data shows that the majority of cultivars licensed in Western nations by law must have a content of less than 0.3% THC (dry weight) in the upper, flowering portion, and in some jurisdictions regulations require less than 0.2%[1,11-12]

L48: etc[17-18].Phenolic

Reply for the review: etc[17-18].Phenolic Among them, phenolic compounds are the main research objects. Including cannabidiol (CBN, cannabinol), cannabidiol (CBD, cannabidiol), cannabidiol (CBDV, Cannabidivarin), etc[17-18].Phenolic substances represented by cannabidiol (CBD) have rich pharmacological effects [19-20].

     Among them, phenolic compounds are the main research objects. Including cannabidiol (CBN, cannabinol), cannabidiol (CBD, cannabidiol), cannabidiol (CBDV, Cannabidivarin)[17-18].Phenolic substances represented by cannabidiol (CBD) have rich pharmacological effects [19-20].

L58: (Figure 3) . (Also L64 and L65)

Reply for the review: (Figure 3) is that Biosynthesis of cannabinoids in Cannabis sativa. The cannabinoids acids are formed in the biosynthetic process, while corresponding decarboxylation products are formed later by decomposition, e.g. under the influence of heat. L58, L64, L65 discuss synthetic cannabinoid and especially the cannabinoids acids are formed in the biosynthetic process.

L65: Therefor, 

Reply for the review: Therefor       However(Therefore)

The introduction is poorly execute, does not contain the essential reason, state of the art of the review, and purpose. 

I conclude that the author did not make serious improvement on the manuscript since many basic problems occur thorough the manuscript. 

 Reply for the review: Cannabis, also known as hemp in China, is a non-toxic hemp variety with THC content of tetrahydrocannabinol less than 0.3%, and is also a widely used hemp variety at home and abroad. cannabis can create value in corresponding fields from seeds, stem cores, flowers, leaves, roots and almost all plant parts, and it is a high strength and high performance characteristic blessing. The introduction introduces the distribution and processing of Cannabis in China, and the significance of cannabinoid research.

Submission Date

04 January 2023

Date of this review

14 Jan 2023 01:28:24

Reviewer 2 Report

I believe that the pertinent, and requested, corrections to the manuscript have been made. For my part I think that the way it is now it can be published.

Author Response

Open Review

English language and style

( ) English very difficult to understand/incomprehensible
( ) Extensive editing of English language and style required
( ) Moderate English changes required
( ) English language and style are fine/minor spell check required
(x) I don't feel qualified to judge about the English language and style

Is the work a significant contribution to the field?

Is the work well organized and comprehensively described?

Is the work scientifically sound and not misleading?

Are there appropriate and adequate references to related and previous work?

Is the English used correct and readable?

Comments and Suggestions for Authors

I believe that the pertinent, and requested, corrections to the manuscript have been made. For my part I think that the way it is now it can be published.

Submission Date

04 January 2023

Date of this review

11 Jan 2023 20:20:08

Reply for the review: Dear reviewer: At present, the number of cannabis varieties popularized in the domestic pharmaceutical industry is small, and the content of cannabinoid is low, only about 1%, which is significantly lower than the international level. In recent years, with the exploitation of the medicinal value of cannabinoid, an upsurge of industrial cannabis has arisen in the world. Ministry of Agriculture and Rural Affairs of the People's Republic of China formed a special research team to investigate the industrial cannabis industry in Yunnan province, and will also conduct research in Heilongjiang province. The industry interprets this research activity as that the state has attached great importance to the industrial cannabis industry and does not rule out the introduction of unified policies in the future. The higher the CBD content of industrial cannabis seeds, the higher the benefit per planting area; from the perspective of CBD extraction, if the CBD content of industrial cannabis is increased from 1% to 3%, the raw material cost per kilogram of the extract will drop by 40%; The content not only directly affects the planting efficiency, but also affects the overall competitiveness of my country's industrial cannabis industry. The prospects for developing high-content CBD seeds are good. Thank you for you review.

Reviewer 3 Report

Review comments to the author

Title: ''Research progress of cannabinoids in Cannabis (Cannabis sativa L.) of China''.

Manuscript ID: molecules-2174095.

The work has been carried out correctly and there is nothing wrong with the results.

Recommendation:

Publish without any further revision.

Author Response

Open Review

English language and style

( ) English very difficult to understand/incomprehensible
( ) Extensive editing of English language and style required
( ) Moderate English changes required
(x) English language and style are fine/minor spell check required
( ) I don't feel qualified to judge about the English language and style

Is the work a significant contribution to the field?

Is the work well organized and comprehensively described?

Is the work scientifically sound and not misleading?

Are there appropriate and adequate references to related and previous work?

Is the English used correct and readable?

Comments and Suggestions for Authors

Review comments to the author

Title: ''Research progress of cannabinoids in Cannabis (Cannabis sativa L.) of China''.

Manuscript ID: molecules-2174095.

The work has been carried out correctly and there is nothing wrong with the results.

Recommendation:

Publish without any further revision.

Submission Date

04 January 2023

Date of this review

14 Jan 2023 10:38:28

 Reply for the review: Dear reviewer: China is the country with the largest area of industrial cannabis cultivation, accounting for about half of the world's total industrial cannabis production, accounting for 25% of the world's total industrial cannabis production. In 2016, China's cannabis production is 77000 tons, and it is estimated that China's cannabis production will reach 103000 tons in 2021. China has successfully obtained gene-edited plants and stable transgenic plants with kanamycin resistance in industrial cannabis for the first time. In recent years, the international scientific community and the capital market have paid increasing attention to the development and application of industrial cannabis. It is worth noting that for a long time, under the influence of policy and industrial guidance, China's industrial cannabis breeding is mainly based on fiber and oil varieties. Thank you for your review.